Formalin-induced pain prolongs sub- to supra-second time estimation in rats

Liu Xinhe
Wang Ning wangn@psych.ac.cn
Wang Jinyan
Luo Fei
1 CAS Key Laboratory of Mental Health, Institute of Psychology, Chinese Academy of Sciences , Beijing , China
2 Department of Psychology, University of Chinese Academy of Sciences , Beijing , China
Doyere Valerie
Electronic publication date: 2021 Mar 2
Publication date: 2021
Volume: 9
Electronic Location ID: e11002
Received 2020 Aug 21; Accepted 2021 Feb 2
Copyright: ©2021 Liu et al.
Copyright year: 2021
Copyright holder: Liu et al.
License: This is an open access article distributed under the terms of the Creative Commons Attribution License, which permits unrestricted use, distribution, reproduction and adaptation in any medium and for any purpose provided that it is properly attributed. For attribution, the original author(s), title, publication source (PeerJ) and either DOI or URL of the article must be cited.
License URL: https://creativecommons.org/licenses/by/4.0/

Keywords: Time perception, Formalin-induced pain, Temporal bisection task, Temporal range

Funding: NNSF (National Natural Science Foundation of China) 31671140 31271092 31970926 CAS Key Laboratory of Mental Health, Institute of Psychology KLMH 2014G01 KLMH2016K02 CAS/SAFEA International Partnership Program for Creative Research Teams Y2CX131003 This work was supported by NNSF (National Natural Science Foundation of China) grants to Ning Wang (31671140), Jin-Yan Wang (31271092), and Fei Luo (31970926), grants from CAS Key Laboratory of Mental Health, Institute of Psychology (KLMH 2014G01, KLMH2016K02), and a grant from the initiation fund of the CAS/SAFEA International Partnership Program for Creative Research Teams (Y2CX131003). The funders had no role in study design, data collection and analysis, decision to publish, or preparation of the manuscript.

==============================
Background

Temporal estimation can be influenced by pain, which is a complex psychological and physiological phenomenon. However, the time range in which perception is most sensitive to pain remains unclear.

Methods

In the present study, we explored the effects of acute inflammatory pain on time perception in the sub- to supra-second (0.6–2.4-s) and supra-second (2–8-s) ranges in rats. Plantar formalin injection was used to induce acute inflammatory pain, and a temporal bisection task was used to measure time perception. Task test sessions were held for five consecutive days (one per day): the day before injection (baseline), immediately after injection, and the three post-injection days. The point of subjective equality (PSE, which reflects the subjective duration) and Weber fraction (which reflects temporal sensitivity) were calculated and analysed.

Results

In the 0.6–2.4-s range, the PSE was significantly lower, indicating prolonged subjective duration, in the formalin group relative to the saline group (p = 0.049) immediately after injection. Formalin-induced pain also tended to lengthened time perception in the 0.6–2.4-s range on post-injection days 2 (p = 0.06) and 3 (p = 0.054). In the 2–8-s range, formalin injection did not affect the PSE or Weber fraction.

Conclusions

The enhanced effect of pain on temporal perception in the sub- to supra-second range is observed in this study and this effect is attenuated with the prolongation of estimated time, even in rats.

Introduction

Pain plays an important role in protecting individuals from potential or further injury (Baliki & Apkarian, 2015). Stress and anxiety often accompany acute pain, which has short-term impacts on attention, motivation and decision making (Wiech, 2016; Wiech & Tracey, 2013). A growing body of evidence suggests that acute pain influences time perception. Human laboratory studies have shown that healthy subjects overestimate the durations of painful stimuli (Khoshnejad et al., 2017; Piovesan et al., 2019) and neutral stimuli paired with painful stimuli (Hare, 1963; Ogden et al., 2015). In the context of acute pain, healthy subjects experienced a prolonged sense of neutral signalling (Ogden et al., 2015; Rey et al., 2017).

Considerable number of evidences suggest that the perception of short but not long duration is modulated by negative emotional stimulus; the greatest effect has been observed for short-term (2-s) time perception, with a gradual decrease with the prolongation of duration (to 4–6 s) (Angrilli et al., 1997; Noulhiane et al., 2007). The reason for this phenomenon may be that emotional stimuli influence time perception from various psychological processes. The classic pacemaker-accumulator model entails the assumption that working memory, reference memory and arousal are involved in time perception (Gibbon, Church & Meck, 1984; Treisman, 1963). Noulhiane et al. (2007) and Treisman (1963) suggested that emotional stimulus–induced high arousal increases the internal clock rate. Some studies have shown that this arousal only affects the processing of short duration (no more than 3–4 s) (Angrilli et al., 1997; Noulhiane et al., 2007).

As a complex physiological and psychological phenomenon with sensory, emotional and cognitive components (Melzack & Wall, 1965), pain has been found to increase physiological arousal levels (Barr, 1998; Santuzzi et al., 2013). Therefore, pain might modulate time perception (or clock) only in short duration. However, a notable problem is most studies of the effect of acute pain on time perception have focused on short (<2.4-s) temporal durations (Liu et al., 2019; Ogden et al., 2015; Rey et al., 2017). Thus, our understanding of time processing in the presence of pain remains limited. Therefore, it is important to clarify the effect of pain on long as well as short temporal range.

In human studies, long temporal ranges are affected easily by strategies such as counting (Thönes & Hecht, 2017). In order to know the effects of pain on longer temporal range, using animal is helpful because animal does not use such strategy. They are thus useful for performing time perception tasks over long temporal ranges (Brown et al., 2011; Kamada & Hata, 2018; Kamada & Hata, 2019; Orduña, Hong & Bouzas, 2011). Temporal bisection task is a classic paradigm of time perception, which is used widely in animals (Brown et al., 2011; Church & Deluty, 1977; Deane et al., 2017; Kamada & Hata, 2018; Kamada & Hata, 2019; Liu et al., 2019; Meck, 1983; Soares, Atallah & Paton, 2016) and humans (Fayolle, Gil & Droit-Volet, 2015; Huang et al., 2018; Rey et al., 2017; Tipples, 2011). In this study, we used the temporal bisection task to investigate how acute inflammatory pain modulates time perception over shorter (sub- to supra-second range, 0.6–2.4 s) and longer (supra-second or second to minute range, 2–8 s; Kamada & Hata, 2018; Kamada & Hata, 2019) temporal ranges in rats. The acute inflammatory pain was induced by subcutaneous injection of formalin into the hind paw, and the rats were trained to perceive temporal durations and classify them as long and short in the temporal bisection task. The experimental data obtained were fitted with a theoretical model, and the point of subjective equality (PSE, a measure of subjective duration) and Weber fraction (a measure of temporal sensitivity) within the temporal ranges examined were calculated. We also explored post-injection effects on time perception, as formalin-induced pain has been shown to influence animals’ emotion-related behaviours even days after injection (Jiang et al., 2014; Johansen, Fields & Manning, 2001).

Materials & Methods

Animals

Thirty-two male Sprague–Dawley rats (weighing 230–250 g on arrival, purchased from Charles River, Beijing, China) were used in this study. The rats were allowed to adapt to the laboratory environment for at least 1 week before the experiment was conducted, and they were handled (by stroking them or letting them move freely on the experimenter’s arms for 5 min) once per day to familiarise them with manipulation by the experimenter. They were housed individually in separate cages with food and water available ad libitum; but since the rats were able to get water from the task, we removed the water bottle from the feeding cage the day before the formal experiment. When a rat did not perform the task on any day, it was allowed to drink freely from a standard 250-ml bottle for 5 min. In addition, we recorded the rats’ body weights every 3–4 days and gave them an additional 5 min access to water when a loss of >10 g from the last recording was noted. During the experimental period, the rats’ body weights were maintained between 270 and 360 g.

The room in which the animals were housed was temperature and humidity controlled (22  ± 2 °C, 50%  ± 10% humidity), with a 12/12-h light/dark cycle (lights on at 8:00 pm). All experiments were performed in accordance with the National Institutes of Health guidelines for the care and use of laboratory animals (ISBN: 0-309-05377-3, First Printing, 1996) and approved by the Institutional Review Board of the Institute of Psychology, Chinese Academy of Sciences. The approve number is H16036. All rats were alive upon completion of this research and were used in preliminary experiments for another study.

Experimental apparatus

Temporal discrimination training and temporal bisection testing of each rat were conducted in the same operant box (21 cm W × 32.5 cm L × 42.5 cm H) located in a soundproof chamber. The box was made of acrylic, had a strip floor and was positioned above a waste catch pan. Two retractable levers (ENV-112CMP; MED Associates Inc., St. Albans, VT, USA) were mounted symmetrically on one wall of the box, 9 cm above the floor. A liquid dispenser (ENV-201A) and water receptacle (2.5 cm above the floor) were spaced equally between the levers to provide water as the reinforcer. As the to-be-timed auditory cue, a pure-tone stimulus (2,900 Hz, 65 dB) was presented by a tone generator (ENV-223AM) mounted on the wall above the liquid dispenser. An illuminated infrared detective nose-poke hole (ENV-114BM) was mounted in the middle of the wall opposite the levers, 2.5 cm above the floor. An indicator lamp (ENV-221M) was located directly above the nose-poke hole. An exhaust fan was mounted on the soundproof chamber. The operant box was connected to a computer that recorded the rats’ experimental events. The experimenter furnished all output and input devices mounted in the box. The box was cleaned with water and medical-grade alcohol after each session.

Experimental procedure

The temporal bisection task was conducted with 16 rats each for the sub- to supra-second and supra-second ranges. After training, each rat was first submitted to three test sessions (one per day) to stabilise its performance. Then, each rat performed five test sessions on five consecutive days (one per day): the day before formalin injection (baseline), immediately after formalin injection, and 1–3 days after formalin injection. Before the day of injection, the rats were assigned to formalin and saline control groups (n = 8 each) according to their baseline performance. All experiments were performed during the rats’ dark cycle.

Temporal discrimination training

Temporal discrimination training was performed using a procedure modified according to previous research (Brasted et al., 1997; Callu et al., 2009; Deane et al., 2017). It consisted of lever-press training (1–2 days), forced-choice training (3 days), and free-choice response training (12 days). In each trial of a lever-press training session (1-h duration, no limit on trial number), an anchor-duration tone was presented, and only the corresponding lever was extended. The anchor-duration tone was the short or long tone used for conditioning (0.6 or 2.4 s for the sub- to supra-second range and 2 or 8 s for the supra-second range). For the forced-choice training trials (n = 100, equal numbers with long and short anchor durations), each tone was triggered by a rat’s nose-poke behaviour. Only the corresponding lever was extended following the termination of the tone. For the free-choice response training trials (n = 100), the tones were triggered by the rats’ nose-poke behaviours, and their termination was followed by the simultaneous extension of both levers. On the first 6 days of free-choice training, the levers were retracted only after a rat provided the correct response; on the next 6 days, the levers were retracted immediately after any response. Correct responses were followed by the provision of drips of water as a reinforcer. Non-response for 5 s resulted in lever retraction and no reward. The criterion for the successful completion of temporal discrimination training was the achievement of correct response rates ≥ 85% on 3 consecutive days. For all training and testing sessions, the rats were allowed to move freely in the operant box.

Temporal bisection testing

During the testing stage of the temporal bisection task, a series of to-be-timed tones of anchor and intermediate durations was presented, and the rats were free to press any of the levers. In the testing sessions, each trial began with the illumination of the indicator light. The to-be-timed tone was initiated by a nose-poke. At the end of the tone, the two levers were extended simultaneously. Lever retraction was prompted by any lever press or non-response for 5 s. The average intertrial intervals (measured from lever retraction to the beginning of the next trial) per rat and session were 20 s (range, 19–21 s) for the 0.6–2.4-s range and 17 s (range, 13–19 s) for the 2–8-s range. One testing session consisted of 140 trials (30 s/trial): 14 trials each with tones of five intermediate durations without reinforcement, and 35 trials each with the two anchor-duration tones with reinforcement. Typically, the rats obtained about 65 reinforcements per testing session. For the sub- to supra-second–range (0.6–2.4-s) experiment, the five intermediate durations were set at 0.756, 0.952, 1.2, 1.51 and 1.9 s. For the supra-second range (2–8-s) experiment, the intermediate durations were set at 2.51, 3.17, 4, 5.04 and 6.35 s. The sequence of these trials was random, but with the constraint of no more than three consecutive intermediate-duration trials.

Establishment of pain model

Acute inflammatory pain was induced by injection of 50 µl 1% formalin solution into the plantar surface of each rat’s hind paw. An equal-volume saline injection was used as the control. After injection, the rats were returned to the operant box to perform the bisection task, and their nociceptive behaviours (paw licking and paw lifting) during the 70-min test sessions were video recorded. The durations of paw lifting and licking, measured by the experimenter in 5-min epochs, and the cumulative duration of nociceptive behaviours in phases I (0–10 min) and II (20–60 min) were calculated for analysis.

Data analysis

Curve fitting

The proportion of “long” responses (PL,calculated by dividing the number of responses to the “long” lever by the total number of responses in trials with the same tone duration) to each to-be-timed tone in each temporal bisection session was used to analyse time perception. Curve fitting and parameter calculation were performed using the Prism software (version 5; GraphPad Software Inc., San Diego, CA, USA). Following previous work (Deane et al., 2017; McClure, Saulsgiver & Wynne, 2005; Ward & Odum, 2007), the final fitting curve is a cumulative Gaussian distribution function: (1) Ft=a+bσ2π∫−∞texp−t−μ22σ2dt,

where F (t) is the PL when the duration is equal to a given sample (t), µ  is the mean, and σ is the standard deviation (representing the slope of the function). Parameters a represents the low asymptote and b represents the range of function. The mean (µ) is also the PSE, defined as a 50% chance that the animal will provide a “long” response (PL = 50%). The PSE reflects the subjectively perceived length of time, with increases and decreases therein interpreted as under- and overestimation, respectively, of the duration (Kamada & Hata, 2018; Kamada & Hata, 2019; Rey et al., 2017). The Weber fraction, an index of temporal sensitivity, was calculated by dividing the standard deviation by the PSE (Crystal, Maxwell & Hohmann, 2003). A decrease in the Weber fraction indicates an increase in temporal sensitivity (Deane et al., 2017; McClure, Saulsgiver & Wynne, 2005; Ward & Odum, 2007).

The R2 statistic was used to quantify the goodness of fit. For each testing session, an individual rat’s PL was fitted with this model to obtain the PSE and Weber fraction (Figs. 1F and 1G, etc.); the average PL of each group of rats was fitted with this model to obtain the best-fitting curve (Figs. 1D and 1E, etc.). When R2 was <0.7 in curve fitting for any individual rat, the PSE and Weber fraction were considered to be inaccurate and were discarded from the analysis.

Figure 1 Effect of formalin injection on temporal bisection behaviours in the 0.6–2.4-s range.

(A–C) Nociceptive behaviours observed during the temporal bisection task (70 min) and in phases I and II after treatment. (D, E) Best-fitting curves for baseline and the injection day, respectively. (F, G) Average PSEs and Weber fractions for the two groups at baseline and on the injection day, respectively. (H) Correlation between PSEs and total durations of nociceptive behaviours on the injection day. *p < 0.05, **p < 0.01, ***p < 0.001 vs. saline group; #p < 0.05 vs. baseline.

Statistical analysis

To assess the post-treatment effect on the PSE, we calculated the mean PSE and Weber fraction, and calculated the change in the PSE relative to baseline using the following formula: (2) Change in PSE=current-session PSE−baseline PSE∕baseline PSE×100%.

Changes in the Weber fraction in the post-injection sessions were calculated using the same formula. The Prism (version 5; GraphPad Software Inc.) and SPSS (version 25; IBM Corporation, Armonk, NY, USA) software packages were used for graph generation and statistical analyses, respectively. Two-way repeated measure analysis of variance (two-way RM ANOVA) followed by a simple main-effect analysis was conducted to analyse the following variables: nociceptive behaviours (group * time); PLs and response latency for each duration in the temporal bisection task [(group * duration) or (day * duration)]; PSE, Weber fraction and omitted trials at baseline and on the injection day (group * day). Independent-sample t test was used to compare the differences of the cumulative duration of paw licking and lifting, the daily correct response rates, and the omitted trials and the changes in PSE/Weber fraction on each post-injection day between the two groups. If these data do not conform to the assumptions of the statistical method, we will use the correction method. The data are expressed as means ± standard errors of the mean, and the significance level was set at α <0.05. Data from trials with no lever-press response were omitted from the analysis.

Results

Effect of formalin-induced pain on temporal estimation in the sub- to supra- second range

As shown in Fig. 1A, formalin injection induced a typical biphasic pattern of nociceptive behaviours. Compared with the saline group, the formalin group exhibited significantly more such behaviours during the temporal bisection task [interaction of treatment and time: F (13, 182) = 8.624, p < 0.001; treatment effect: F (1, 14) = 28.500, p < 0.001; time effect: F (13, 182) = 10.844, p < 0.001; Fig. 1A]. Cumulative duration of paw licking and lifting after formalin injection were increased significantly in phases I [t (14) = −5.711, p < 0.001; Fig. 1B] and II [t (14) = −4.433, p < 0.001; Fig. 1C] compared with those after saline injection.

We compared the correct response rates to anchor-duration tones between two groups at baseline (0.6 s: saline 96.75% ± 1.38%, formalin 96.07% ± 1.20%; 2.4 s: saline 94.29% ± 2.96%, formalin 95.34% ± 3.10%) and on the injection day (0.6 s: saline 90.71% ± 3.23%, formalin 90.57% ± 3.19%; 2.4 s: saline 96.01% ± 2.21%, formalin 97.37% ± 1.05%), respectively. Independent-sample t tests revealed no difference in the correct response rates at baseline or on the injection day, indicating that formalin treatment did not affect the animals’ temporal discrimination ability.

The effect of formalin injection on PLs of temporal bisection task was presented in Figs. 1D and 1E. Two-way RM ANOVA revealed no significant interaction between duration and treatment affecting the PLs at baseline [F (6, 84) = 0.489, p = 0.815, Fig. 1D] and on the injection day [F (6, 84) = 2.486, p = 0.23, Fig. 1E]. Nevertheless, the significant duration effect showed that the rats could discriminate the durations of the to-be-timed tones (p < 0.001). For each group, the differences of PLs between baseline day and injection day were also compared by Two-way RM ANOVA. A significant interaction of the day and duration was found in the formalin group [F (6, 42) = 2.486, p = 0.038] and a simple main-effect analysis showed that PLs on the injection day were higher than these at baseline at 0.952 s (p = 0.030) and 1.2 s (p = 0.017). No such effect was observed in the saline group [F (6, 42) = 0.790, p = 0.583]. R2 values for curve fitting ranged from 0.86 to 0.99 (average: 0.94) in the two groups at baseline and on the injection day, with no difference between groups.

As presented in Fig. 1F, two-way RM ANOVA revealed significant interaction of the treatment and day affecting PSE values [F (1, 14) = 6.405, p = 0.024]. A simple main-effect analysis showed that the PSE was lower in the formalin group than in the saline group on the injection day [F (1, 14) = 4.67, p = 0.049], but not at baseline [F (1, 14) = 0.02, p = 0.899]. The Weber fraction, however, did not change significantly [interaction of treatment and day: F (1, 14) = 0.017, p = 0.897, Fig. 1G]. Moreover, we found a negative correlation between the total duration of nociceptive behaviours and the PSE on the injection day in all rats (r = −0.505, p = 0.046, Pearson product-moment correlation; Fig. 1H).

We also compared the omitted trials at baseline (saline, 4.125 ± 3.199; formalin, 0.875 ± 0.372) and on the injection day (saline, 4.250 ± 2.296; formalin, 3.750 ± 0.862). Two-way RM ANOVA revealed that the treatment had no significant effect in the omitted trials [treatment effect: F (1, 14) = 0.400, p = 0.537; interaction of day and treatment: F (1, 14) = 3.264, p = 0.092]. The response latency (defined as the time between lever extension and pressing) did not differ between groups at baseline [treatment effect: F (1, 14) = 2.140, p = 0.166; interaction of duration and treatment: F (6, 84) = 1.814, p = 0.106, Fig. S1A] or on the injection day [treatment effect: F (1, 14) = 0.113, p = 0.741; interaction of duration and treatment: F (6, 84) = 0.693, p = 0.656; Fig. S1B].

Effect of formalin-induced pain on temporal estimation in the supra-second range

In this range, formalin injection also induced a typical biphasic pattern of nociceptive behaviours during the temporal bisection session. As shown in Fig. 2A, formalin injection significantly increased nociceptive behaviours compared with saline injection [treatment effect: F (1, 13) = 26.183, p <0.001; time effect: F (13,169) = 11.190, p < 0.001; interaction of treatment and time: F (13,169) = 7.363, p < 0.001]. The cumulative duration of paw licking and paw lifting was significantly greater in the formalin group than in the saline group in phases I [t (13) = −6.577, p < 0.001; Fig. 2B] and II [t (13) = −3.155, p = 0.019; Fig. 2C].

Figure 2 Effect of formalin injection on temporal bisection task performance in the 2–8-s range.

(A–C) Nociceptive behaviours observed during the temporal bisection task (70 min)and in phases I and II after treatment (D, E) Best-fitting curves for baseline and the injection day, respectively. (F, G) Average PSEs and Weber fractions for the two groups at baseline and on the injection day, respectively. (H) Correlation between PSEs and total durations of nociceptive behaviours on the injection day. *p < 0.05, **p < 0.01, ***p < 0.001 vs. saline group.

Correct response rates to anchor durations was also analysed. Independent-sample t tests revealed no significant difference at baseline (2 s: saline 99.28% ± 0.47%, formalin 100.00% ± 0.00%; 8 s: saline 97.99% ± 1.05%, formalin 94.67% ± 2.29%) or on the injection day (2 s: saline 99.62% ± 0.38%, formalin 95.81% ± 3.74%; 8 s: saline 94.52% ± 1.13%, formalin 97.53% ± 1.16%), indicating that formalin treatment did not influence the temporal discrimination ability within this temporal range.

The effect of formalin injection on PLs in the supra-second range was presented in Figs. 2D and 2E. Two-way RM ANOVA revealed no significant interaction of duration and treatment affecting the PL at baseline [F (6, 78) = 0.405, p = 0.874] or on the injection day [F (6, 78) = 0.680, p = 0.666]. However, the treatment had a significant main effect on the injection-day PL[F (1, 13) = 5.186, p = 0.040]; no such effect was observed at baseline [F (1, 13) = 0.202, p = 0.660]. The significant duration effect revealed that the rats discriminated the durations of the to-be-timed tones (p < 0.001). For each group, two-way RM ANOVA was also used to compare the differences of PLs between baseline day (Fig. 2D) and injection day (Fig. 2E). No significant interaction of the day and duration was found in the formalin group [F (6, 36) = 0.804, p = 0.573] or saline group [F (6, 42) = 0.335, p = 0.915]. R2 values for curving fitting ranged from 0.81 to 0.99 (average: 0.95) in the two groups at baseline and on the injection day, with no difference between groups [data from one rat were excluded because the R2 values from all tests were low (0.734 ± 0.168)].

As shown in Figs. 2F and 2G, no significant interaction of the treatment and day affecting the PSE [F (1, 13) = 0.772, p = 0.396] or the Weber fraction [F (1, 13) = 0.892, p = 0.379; Fig. 2G] was observed. In addition, we observed no correlation between the PSE and nociceptive behaviours of all rats in the supra-second range (r = −0.044, p = 0.877, Pearson product-moment correlation; Fig. 2H).

Two-way ANOVA revealed no significant difference between omitted trials at baseline (saline, 4.571 ± 1.45; formalin, 10.875 ± 3.966) and on the injection day [saline, 7.571 ± 2.203; formalin, 10.500 ± 3.375; treatment effect: F (1, 13) = 1.199, p = 0.293; interaction of treatment and day: F (1, 13) = 1.503, p = 0.242]. At baseline, response latency did not differ between groups [treatment: F (1, 13) = 0.848, p = 0.374; interaction of duration and treatment: F (6, 78) = 1.120, p = 0.358; Fig. S2A]. A significant interaction of duration and treatment affected response latency was observed on the injection day [F (6, 78) = 4.169, p = 0.001]; no such interaction was observed at baseline [F (6, 78) = 0.418, p = 0.865; Fig. S2B]. A simple main-effect analysis showed that the formalin treatment prolonged the response latency in the 4-s trial (p = 0.035).

Post-treatment effect of formalin-induced pain on temporal estimation

The correct response rates of temporal bisection tasks performed in the 0.6–2.4-s range on post-injection days were analysed. Independent-sample t tests revealed no significant difference in the correct response rates to anchor-duration tones on post-injection day 1 (0.6 s: saline 94.94% ± 1.99%, formalin 93.51% ± 3.42%; 2.4 s: saline 95.89% ± 2.22%, formalin 95.27% ± 1.32%), 2 (0.6 s: saline 95.63% ± 1.46%, formalin: 95.56% ± 2.61%; 2.4 s: saline 96.05% ± 0.93%, formalin 96.04% ± 1.70%), or 3 (0.6 s: saline 96.78% ± 1.66%, formalin 97.82% ± 1.44%; 2.4 s: saline 98.21% ± 0.93%, formalin 96.07% ± 1.20%). These results indicate that the post-injection effect of formalin treatment did not influence the animals’ temporal discrimination ability for the anchor durations.

The results of temporal bisection tasks performed in the 0.6–2.4-s range on post-injection days are presented in Figs. 3A–3C. No significant interaction of treatment and duration affecting the PL was observed [post-injection day 1: F (6, 84) = 1.861, p = 0.097; post-injection day 2: F (6, 84) = 1.025, p = 0.415; post-injection day 3: F (6, 84) = 1.706, p = 0.130]. The main treatment effect also was not significant [post-injection day 1: F (1, 14) = 1.304, p = 0.273; post-injection day 2: F (1, 14) = 2.300, p = 0.152; post-injection day 3: F (1, 14) = 1.359, p = 0.263]. The main effect of duration was significant in all tests [post-injection day 1: F (6, 84) = 102.703, p <0.001; post-injection day 2: F (6, 84) = 92.340, p < 0.001; post-injection day 3: F (6, 84) = 116.652, p < 0.001], indicating that the rats could discriminate the durations of the to-be-timed auditory stimuli. R2 values for curve fitting ranged from 0.87 to 1 (average: 0.96), with no significant difference between groups on any of the 3 days.

Figure 3 Post-injection effect of formalin treatment on temporal bisection task performance in the 0.6–2.4-s and 2–8-s ranges.

(A–C) Average PLs for to-be-timed tone durations in the 0.6–2.4-s range and best-fitting curves for post-injection days 1–3, respectively. (D–F) Results of temporal bisection tasks in the 2–8-s range on post-injection days 1–3, respectively.

As shown in Figs. 4A and 4B, we compare the change in the PSE and Weber fraction in the 0.6–2.4-s range from baseline on each post-injection day (Original PSE, day 1: saline 1345.375 ± 53.468, formalin 1183.15 ± 111.639; day 2: saline 1413.375 ± 124.251, formalin 1140.388 ± 128.744; day 3: saline 1435.875 ± 82.933, formalin 1221.438 ± 107.257; Original Weber fraction, day 1: saline 0.217 ± 0.059, formalin 0.222 ± 0.054; day 2: saline 0.231 ± 0.063, formalin 0.254 ± 0.038; day 3: saline 0.231 ± 0.048, formalin 0.222 ± 0.032). Independent-sample t tests showed that the formalin treatment tended to reduce the change of PSE on post-injection days 2 [t (9.353) = 2.143, p = 0.060] and 3 [t (8.527) = 2.238, p = 0.054], see Fig. 4A.

Figure 4 Changes in PSE and Weber fraction from baseline on post-injection days 1–3.

Average changes in PSE and Weber fraction for temporal tasks in the 0.6–2.4-s (A, B) and 2–8-s (C, D) ranges.

Omitted trials from the three post-days (day 1: saline 6.375 ± 3.201, formalin 3.625 ± 1.992; day 2: saline 5.000 ± 3.082, formalin: 2.500 ± 0.810; day 3: saline 5.250 ± 3.521, formalin 1.625 ± 0.431) were also analysed. Independent-sample t tests revealed no post-effect of formalin treatment in the omitted trials from any day. Two-way RM ANOVA (duration * treatment) revealed that the treatment had no significant effect on response latency on any of the 3 days [treatment effect: F (1, 14) ≤ 1.924, p ≥ 0.187; interaction of duration and treatment: F (6, 84) ≤ 1.862, p ≥ 0.097, Fig. S3].

In the 2–8-s range, the correct response rates to anchor-duration tones did not differ significantly on post-injection day 1 (2 s: saline 100.00% ± 0.00%, formalin 97.96% ± 1.20%; 8 s: saline 94.57% ± 1.67%, formalin 95.45% ± 1.97%), 2 (2 s: saline 99.255 ± 0.49%, formalin 100.00% ± 0.00%; 8 s: saline 97.14% ± 2.33%, formalin 96.73% ± 2.01%), or 3 (2 s: saline 100.00% ± 0.00%, formalin 98.78% ± 0.85%; 8 s: saline 93.77% ± 2.17%, formalin 95.00% ± 2.27%). These results indicate that the formalin treatment did not influence the animals’ temporal discrimination ability for the anchor durations.

The results of temporal bisection tasks performed in the 2–8-s range on post-injection days are shown in Figs. 3D–3F. Two-way RM ANOVA revealed no significant interaction of treatment and duration affecting the PL on any day [post-injection day 1: F (6, 78) = 0.303, p = 0.934; post-injection day 2: F (6, 78) = 0.700, p = 0.651; post-injection day 3: F (6, 78) = 0.695, p = 0.654]. The effect of treatment on PL also was not significant [post-injection day 1: F (1, 13) = 0.097, p = 0.761; post-injection day 2: F (1, 13) = 0.265, p = 0.615; post-injection day 3: F (1, 13) = 0.331, p = 0.575]. Duration had a significant effect on the PL in all tests (p <0.001). R2 values for curve fitting ranged from 0.83 to 1 [average: 0.95; the PSE and Weber fraction from one rat were excluded because the R2 values were low (average: 0.6) on post-injection day 2], with no significant difference between groups on any of the 3 days.

As shown in Figs. 4C and 4D, the treatment had no significant effect on the change of PSE or Weber fraction from baseline on any of these post-injection day (Original PSE, day 1: saline 3.427 ± 0.216, formalin 3.403 ± 0.224; day 2: saline 3.486 ± 0.116, formalin 3.784 ± 0.23; day 3: saline 3.500 ± 0.180, formalin 3.800 ± 0.419; Original Weber fraction , day 1: saline 0.191 ± 0.026, formalin 0.185 ± 0.043; day 2: saline 0.130 ± 0.029, formalin 0.207 ± 0.042; day 3: saline 0.186 ± 0.038, formalin 0.260 ± 0.056).

Omitted trials from the three post-injection days (day 1: saline 6.750 ± 4.259, formalin 13.143 ± 5.501; day 2: saline 7.875 ± 4.673, formalin 9.857 ± 4.752; day 3: saline 8.125 ± 4.525, formalin 11.000 ± 3.9) were analysed. Independent-sample t tests revealed no effect of the formalin treatment on omitted trials from any day. The response latency did not differ significantly on post-injection days 1 and 2 [treatment: F (1, 13) ≤ 2.58, p ≥ 0.132; interaction of duration and treatment: F (6, 78) ≤ 1.365, p ≥ 0.239; Fig. S4]. Interaction of treatment and duration affecting response latency was observed for post-injection day 3 [treatment: F (1, 14) = 0.913, p = 0.357; interaction of duration and treatment: F (6, 78) = 2.666, p = 0.021; Fig. S4C]. A simple main-effect analysis revealed significantly more response latency in the 4-s trials in the formalin group (p = 0.022).

Discussion

In this study, we investigated the effect of formalin-induced pain on time estimation at two scales in rats. Plantar formalin injection lengthened subjective time perception on the injection and post-injection days in the sub–to supra-second temporal range (0.6–2.4 s), but not in the supra-second range (2–8 s).

Human and animal studies exploring how acute pain modulates time perception within relatively short temporal ranges have yielded similar results. In humans, lengthened temporal estimation of neutral stimuli has been observed with laser-induced heat pain (0.35–1.37-s stimulus duration) (Ogden et al., 2015), in a cold pressor test (0.25–0.75-s duration) (Rey et al., 2017), with electro-cutaneous stimuli rated as inducing a high level of pain (0.24–1.30-s duration) (Piovesan et al., 2019), and in children with migraine (1.5–1.9-s duration) (Vicario et al., 2014). In rats, formalin-induced pain has been shown to increase temporal perception during temporal bisection task with to-be-timed tones in the 1.2–2.4-s range (Liu et al., 2019). Generally, the results of the present study are consistent with previous findings: acute pain lengthened time perception over short (<2.4-s) durations. We found that formalin injection into the hind paw induced more nociceptive behaviours during the whole task period (70 min), suggesting that the animals felt pain while performing the task. This result is in agreement with previous findings that subcutaneous formalin injection induced obvious and persistent nociceptive behaviours (Jiang et al., 2014; Johansen, Fields & Manning, 2001). We observed a negative relationship between the duration of nociceptive behaviour and the PSE, an index of subjective duration, indicating that subjective duration lengthened with increased nociceptive behaviour. Pain is believed to influence psychological processes such as motivation (Schwartz et al., 2014) and attention (Freitas et al., 2015), which are related to temporal bisection task performance. In the sessions conducted with to-be-timed tones in the 0.6–2.4-s range, we found no difference in the number of omitted trials (a measure of motivation) or the response latency (a measure of attention). These results indicate that the rats’ motivation and attention remained normal after formalin injection. Thus, we ruled out these factors as the cause of temporal overestimation in the 0.6–2.4-s range. This perceptual effect for short-duration has been related to high arousal (self-reported and quantified by skin conductance and heart rate) in humans (Angrilli et al., 1997; Mella, Conty & Pouthas, 2011). Thus, acute pain–induced high arousal may mediate temporal overestimation in this duration range.

Few studies have examined the effect of pain on temporal perception over long durations, perhaps because the perceptual process is more complex than for short durations. Subjects must focus their attention on to-be-timed stimuli for a long time, necessitating greater involvement of cognitive abilities such as selective and sustained attention and working memory (Droit-Volet, 2013; Fraisse, 1984). In addition, human subjects may adopt strategies such as counting to estimate long stimulus durations (Thönes & Hecht, 2017), meaning that the process is not purely perceptual. Healthy subjects who were required to put their hands in cold (7 °C) water (pain condition) or warm (35 °C) water (control condition) for 120 or 300 s retrospectively underestimated these durations of pain relative to controls (Hellstrom & Carlsson, 1997; Thorn & Hansell, 1993). In contrast, Khoshnejad et al. (2017) found that healthy subjects reported longer subjective durations of a high-intensity thermal stimulus (rated as high pain, 10–11 s) than of a low-intensity thermal stimulus (rated as low pain). The 2–8-s range is used commonly in animal studies of time perception. Using the temporal bisection task in this range, Kamada and Hata (Kamada & Hata, 2018; Kamada & Hata, 2019) found that rats underestimated the duration of an auditory cue paired with an electric foot shock (0.4 mA), whereas Meck (1983) observed that an electric foot shock (0.2 mA) induced temporal overestimation in rats. The intensities of the electrical stimuli used in those studies may not have been sufficient to induce nociception, and the authors did not provide nociceptive data. Overall, the effect of pain on time perception in this range remains unclear.

In the 2–8-s range, we found that PLs to intermediate-duration tones were greater in the formalin group than in the saline group, with no significant difference in the PSE. These results indicate that formalin-induced pain had a weaker effect in this range relative to the 0.6–2.4-s range. In addition, pain is known to automatically attract attention (Baliki & Apkarian, 2015). Nociceptive stimulus–induced pain has been shown to disrupt attention-based cognitive tasks in animals (Boyette-Davis, Thompson & Fuchs, 2008; Freitas et al., 2015; Pais-Vieira et al., 2012), and the distraction of attention from to-be-timed stimuli has been found to shorten subjective stimulus durations in humans (Casini & Macar, 1997; Chen & O’Neill, 2001; Macar, Grondin & Casini, 1994) and animals (Buhusi & Meck, 2006; Buhusi & Meck, 2009). In addition, we found that formalin treatment increased the response latency in intermediate-duration trials (4 s) in this range, but not in the 0.6–2.4-s range. Theoretically, animals perceive long-duration tones by focusing their attention on them for longer periods. Thus, we believe that the distraction caused by pain interferes with attention during time processing, leading to the prolongation of response latency. The influence of pain may differ according to the stimulus duration; the perception of short-duration may not require the involvement of attention and could be lengthened by pain-induced high arousal, whereas the perception of long-duration may be affected jointly by such arousal and attentional distraction, leading to a weaker lengthening effect in the 2–8-s range.

A post-injection effect of formalin-induced pain, namely the overestimation of tone duration, was observed only in the 0.6–2.4-s range on the second and third days after injection. We suggest that the affective component of formalin-induced pain contributed to the observed post-injection effect. Minor change was also observed after injection of normal saline. Normal saline injection only produced a transient and very mild nociceptive responses (less than 5 min) and did not affect the temporal bisection task on the injection day. Therefore, these less significant changes in saline group may be due to training effects or random fluctuations. Subcutaneous formalin injection into the rat hind paw is a classic method for the creation of a model of acute inflammatory pain. Low-dose (e.g., 1%) formalin induces phase-I and phase-II nociceptive behaviours that do not last until the following day (Fu, Light & Maixner, 2000; Johnston et al., 2012). However, formalin-induced pain can induce conditioned place avoidance within a few days in rats, suggesting the induction of pain-related aversion (Jiang et al., 2014; Johansen, Fields & Manning, 2001). Aversion and fear can induce temporal overestimation (Buetti & Lleras, 2012; Gil & Droit-Volet, 2009; Gil & Droit-Volet, 2012; Grommet et al., 2011; Watts & Sharrock, 1984), and pain-related aversion may have a similar effect. Emotional stimuli have been shown to have strong effects on perception of shorter-duration (ca. 2-s), but not longer-duration (4–6 s) (Angrilli et al., 1997; Noulhiane et al., 2007). These studies support our explanation that the overestimation of the 0.6–2.4s range after formalin injection may be due to emotional dimension of pain. We did not observe the significant post-injection effect of formalin-induced pain in 2–8-s range. However, other studies have shown that conditioned fear cues induce temporal underestimation in the 2–8-s range (Kamada & Hata, 2018; Kamada & Hata, 2019).

Formalin-induced acute pain did not affect temporal sensitivity in the sub–supra-second or supra-second range in this study. Previous research has revealed a negative relationship between the level of chronic stress and temporal sensitivity (measured with a temporal bisection task) in healthy human subjects (Yao et al., 2015). Acute psychosocial stress may induce a stronger cortisol response, but may not significantly affect temporal sensitivity (Yao et al., 2016). Decreased temporal sensitivity has been observed in patients with psychiatric disorders, such as Parkinson’s disease and schizophrenia (Alústiza et al., 2016; Harrington et al., 2011; Jahanshahi et al., 2010), suggesting an association with pathological changes in the brain. Our model of formalin-induced acute pain may not have been sufficient to induce pathological changes in brain function.

Our results show that formalin-induced pain affected time perception in the sub- to supra-second range on days after formalin injection, but we conducted observations on only three such days. Thus, the duration of effect persistence remains unclear. Previous studies have shown that animals can learn to be unresponsive to intermediate-duration after repeated performance of the temporal bisection task (Brown et al., 2011). To avoid the omission of an excessive number of trials, we measured time perception in only five consecutive test sessions. Despite this limitation, the present study can provide guidance for further research. For example, combining formalin pain induced conditioned placed aversion paradigm with time perception task can obtain pain emotion related behavioral results and time perception data at the same time, which will clarify the impact of pain emotion on time perception.

This study explored the optimal time window for pain to affect time perception, and partially confirmed the possible role of pain emotion in it. These results can provide evidence for some theoretical models of time perception. Striatal beat frequency (SBF) model emphasizes the critical role of cortical activities in temporal information processing (Van Rijn, Gu & Meck, 2014). As a multidimensional and complex experience, pain can activate many brain regions, including emotion related brain areas, such as anterior cingulate cortex and anterior insular; sensory related brain areas, such as primary somatosensory cortex and lateral thalamus; cognitive related brain areas, such as medial prefrontal cortex (Wiech, 2016). The medial prefrontal cortex has been shown to play a key role in the encoding of temporal information (Emmons et al., 2017; Xu et al., 2014). Although the influence of other pain related cortical regions on time perception coding remains to be explored, we can still believe that exploring the impact of pain on time perception and its mechanism will help support the SBF model and even help to establish a more optimized physiological model of time perception.

Conclusions

This study showed that the effect of acute pain on time perception depends on the duration of the tone to be estimated. Acute pain leads to more significant overestimation of shorter-duration than of longer-duration. Pain-induced high arousal and distraction may jointly affect time perception, and this effect may differ between the short-term and long-term ranges. In addition, the emotional component of pain may play a role independent of the sensory component. In future research, the effects of the emotional processing of pain on temporal perception should be explored further.

Supplemental Information

Data S1 Raw data

Click here for additional data file.

Supplemental Information 1 Author checklist

Click here for additional data file.

Figure S1 Response latency to each to-be-timed tone durations in baseline (A) and injection (B) sessions in 0.6 –2.4s range

There is no significant difference between formalin group and saline group in each session.

Click here for additional data file.

Figure S2 Response latency to each to-be-timed tone durations in baseline (A) and injection (B) sessions in 0.6 –2.4s range

Formalin treatment prolonged response latency to 4-s trials and 5.04-s trials in the injection test (B). *p < 0.05, **p < 0.01 vs. saline group.

Click here for additional data file.

Figure S3 Response latency to each to-be-timed tone durations in three days (A) –(C) after formalin treatment in 0.6 –2.4s range

There is no significant difference between formalin group and saline in each session.

Click here for additional data file.

Figure S4 Response latency to each to-be-timed tone durations in three days (A) –(C) after formalin treatment in 2 –8s range

Formalin treatment prolonged the response latency to 4-s trials in post-injection day3 (C). *p < 0.05 vs. saline group.

Click here for additional data file.

Additional Information and Declarations

Competing Interests

Author Contributions

Animal Ethics

Data Availability

The authors declare there are no competing interests.

Xinhe Liu conceived and designed the experiments, performed the experiments, analyzed the data, prepared figures and/or tables, authored or reviewed drafts of the paper, and approved the final draft.

Ning Wang conceived and designed the experiments, prepared figures and/or tables, authored or reviewed drafts of the paper, and approved the final draft.

Jinyan Wang and Fei Luo conceived and designed the experiments, authored or reviewed drafts of the paper, and approved the final draft.

The following information was supplied relating to ethical approvals (i.e., approving body and any reference numbers):

The Institutional Review Board of the Institute of Psychology, Chinese Academy of Sciences approved this research. The approval number is H16036.

The following information was supplied regarding data availability:

Raw data are available in the Supplemental Files.

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
