# Peer review of "Formalin-induced pain prolongs sub- to supra-second time estimation in rats"

_PeerJ, doi:10.7717/peerj.11002_

## Round 0.1 · original submission · Major Revisions

I do agree with both reviewers that the manuscript needs some amount of work, in particular in the results section and statistical inferences. I will not reiterate every suggestion made by the reviewers, as in my view they are all valid. In addition, I wonder whether ANOVAs are still valid statistics when big differences in variability seem to exist between saline and formalin (e.g., figure 4D). It may also be worth discussing whether the significant differences obtained at a specific time between saline and formalin may be due to changes in saline more than in formalin (figure 3D). Finally, I suggest you take advantage of a professional English editing service familiar with scientific matters, as some sentences are quite difficult to follow.

Both reviewers have attached documents with their review, reviewer 1 for help in statistical matters, reviewer 2 for an in-depth formal review. Please respond point-by-point to that formal review carefully.

·

Basic reporting

No comment.

Experimental design

No comment.

Validity of the findings

No comment.

Additional comments

The journal requires to submit ARRIVE 2.0 checklist as a supplemental file.
https://peerj.com/about/policies-and-procedures/#animal-research
Please check and submit it.

Follows are point to point comments.

L(ine). 72
The study by Meck (1983) should be deleted here, because the study was for rats.

L. 73
"Involvement of different mechanisms in time perception" might be "Involvement of different mechanisms *in the modulation of* time perception". The assumed mechanism underlying the time perception is SET (scalar expectancy theory) or information processing theory by Gibbon (1984) in both short and long duration, isn't it? So, the "mechanism itself" is same in both durations. The difference is "in short duration, but not in long duration, the arousal effects on internal clock". Right?
Similarly, in line 78, the mean of "from" different aspects is unclear. What author's want to say is, as above, in short duration, but not in long duration, the arousal effects on internal clock"?
Taken together, the topic (first) sentence of the second paragraph might be such as "Considerable number of evidences suggest that the perception of short but not long duration is modulated by negative emotional stimulus" This is a phenomenon. After that, the description moves onto the mechanism, namely, the arousal modulates clock only in long duration.

L. 79
"attention and arousal are involved" is seems to be false. The Gibbon's model did not include these components and not refer to the effect of arousal or attention. Alternatively, Treisman (1963) --- Psychological Monographs: Gneral and Applied, 77, 1-31. --- refers to it. Treisman is more suitable here because the model by Gibbon is basically based on animal studies, whereas the Treisman's one is a traditional model for human timing and refers to the effect of arousal on pacemaker. So, bothe of Treisman and Gibbon should be cited for stating that the accumulator-pacemaker model, and Treisman and Noulhiane et al. are cited for stating the effect of arousal.

L. 86
One piece of logic seems to be skipped between the sentence "As a complex ... Freitas et al 2015)" and "A notable ...". The former paragraph said that arousal modulates clock only in short duration. The first sentence of this paragraph said that pain increases arousal. So the missing piece seems to be "Therefore, pain might modulate time perception (or clock) only in short duration." After that, "However, a notable problem is ...". I believe this revision will fulfill the chain of logic.

L. 89
If the paragraph is finished such as "Therefore, it is important to clarify the effect of pain on long as well as short temporal range", the connection to the next paragraph seems to be smooth.

L. 90
Again, the chain of logic between the first and second sentence is unclear. The first sentence means "human use counting in long temporal range". In natural way, "by contrast" to this, "the animal did not use counting". Nevertheless, the manuscript said "animals are often trained to perform the time perception task in a long temporal range". The sentence such as "In order to know the effects of arousal on longer temporal range, using animal is helpful because animal does not use such strategy" is needed.

L. 108
Please clarify the procedure of the handling (e.g., duration per day, number of times).

L. 110
Please clarify the extent of water deprivation. That is important information for animal welfare.

L. 115
You have declared the approved No. is H16036 on the on-line check at the submission of the manuscript and confirmed it. You uploaded the approval document as a supplemental file, but I could not find the number on it. Could you evidence the approved No. of this experiment is actually H16036? If you can evidence it, please show the No. in Method (The guidance require it, if the No. is available).

L. 117-123
The section of "Temporal bisection task" consists of only two sentences and contains not so much information.
I recommend delete this section and that the first sentence is moved onto the introduction (around L. 96?) and the second one to the procedure.

L. 119
I recommend to cite the first study using temporal bisection task too (Church & Deluty, 1977, J Exp Psychol Anim Behav Process, 3, 216-28. doi: 10.1037//0097-7403.3.3.216.)


L. 124
Please enlarge the font size of the heading "Experimental apparatus" as the other heading such as "Animals".

L. 139
Although the term "training" is distinguished from "testing" in line 125, "test" is used to mean "session" in line 139. It is confusing. Please replace "test" in line 139 with "session".

L.168
The heading of "Experimental procedure" should be moved after the section of "Experimental apparatus". After that, the Headings of "Temporal discrimination training" and "Temporal bisection testing" should be put on the layer under the heading of "Experimental procedure" (namely, without new line), in order to clarify the flow of the experiment.

L. 144
Please explain the "anchor-duration".

L. around 144
Please clarify the ratio of short:long trials. Maybe 1:1?

L. 160
Please clarify not only the average but also the range of ITI.

L. 179
Is the number of paw licking and paw lifting counted within the drug-injected bisection task? With video-recording or real-time counting? The count was done with blinded? Please clarify.

L. 184
Please clarify the mean of "long response (PL)".

L. 185
Temporal bisection "task" is temporal bisection "session"? Please clarify that the curves were fitted to both individual rats (for analysis of PSE and WF, Fig1 F, G etc) and averaged psychophysical function (Fig.1D, E, etc).

L. 201
Here, change in the PSE and Weber fraction (WF) only were referred. Not only them, mean PSE and WF were calculated. Please describe them too.

L. 208
Many studies claim that Duncan's method is not valid as a multiple comparison (e.g., Hsu, 1996 attached file, etc.). Moreover, as far as I know, the standard way as a post-hoc analysis after the interaction is significant in ANOVA is test for simple main effect, lower parts of ANOVA (e.g., attached file). Please consider to use the simple main effect as a post-hoc analysis.

L. 209
0.05 (5%) is a criteria for significance. P is the calculated value from data. So α=.05, not p=.05.

L. 209
"Temporal bisection task trials with ..." is "The data from trials with..."?

L. 220
Please check the data of Fig. 1C. I calculate them and got 123.00 and 13.63 for formalin and saline, respectively. The graph looks unmatched to the values. Also, please check t value. I got 4.433.

L. 222
"goodness of fit" is more generalized term, so please use "R^2" or "coefficient of determination" here.

L. 222
0.94 is group mean? or minimum value? Please clarify.

L. 223
When is the "two days"? Baseline and drug challenge days? Pleases clarify.

L. 228
"Accuracy" has a specific mean in a context of timing study (that is, the discrepancy between the duration of to-be-timed stimulus and estimated "subjective" time or the length of the subjective time itself). Please replace it with "correct response rate".

L. 232-242
Please rewrite these parts. The Fig. 1D, E is separated by days (baseline and injection) and compare saline vs. formalin in each day. However, the text compares between days in each group. The reason why the control (saline) group is prepared is to compare it with experimental group. So, the important comparison is between groups. Therefor, two-way anova should be done with group by duration in each day. I think three-way anova is not so important here (in line 223). Do you delete it? For PSE and WF, post-hoc analysis should be done with simple main effects, if the interaction of two-way ANOVA (day vs. group) was significant, as mentioned above.

L. 315
Please write the values of statistical tests also for no significant difference in three days.

L. xx
In discussion, please refer to the persistence of pain throughout the session using Fig. 1A-C and Fig. 2A-C.

L. 351
The authors say that "believed to be ... motivation, attention, and decision-making" are influenced by pain, but insist that in this experiment, "motivation, the ability to discriminate these anchor duration and physical activity or attention were still normal" ... "we ruled out these factors". As the authors say, motivation and attention might be ruled out. However, how about "decision-making"? In the latter sentences, the term "decision-making" disappeared. Instead, "ability to discriminate these anchor duration", "physical activity" are introduced. Please clarify the relationship between "decision-making" and "ability to discriminate these anchor duration", "physical activity".

L. 384
What is the "distinct physiological characteristics"? In the manuscript, have the distinct physiological characteristics between short and long duration been referred until here?. Also after this sentence, cognitive but not physiological explanation starts. Could you delete this sentence (, which may reflect ... stimuli.)?

L. 389
Should the animal studies addressing that attention modulates time perception be cited? For example, Buhusi & Meck, 2006, J.Exp. Psy. Animal. Behav. Proc. 32, 329-338.; Buhisi & Meck, (2009) Phl. Trans. Royal. Soc. B,364, 1875-1885.) etc.

L. 389-391
What for is the evidence of the increase in latency here? Please clarify.

L. 396
Please give more detailed explanation for the term "pure".

L. 409
Kamada & Hata (2018; 2019) repeatedly reported that conditioned fear induced underestimation of duration. If you have some ideas to bridge this discrepancy, please mention it. If no idea, at least, please cite them like " ... , but see Kamada & Hata, 2018; 2019."

L. 424
What will occur if you continue the after-injection sessions? You implicates that the effect is conditioned response (CR) around line 407-409. If so, it is expected that prolonged after-injection training extinguish the CR. The last part of this paragraph seems to finish suddenly. I believe that the paragraph can be finished neatly If you add what will occur when the prolonged extinction sessions are given.

Caption of Figure 1
"Fitting curves for the proportions of ..." is "mean and their fitting curves"? (also the caption for Fig. 2).


That's all.

Reviewer 2 ·

Basic reporting

Overall, the paper is clear and unambiguous. They have sufficient references and background context. The article structure is professional, and self-contained.

There are a few small exceptions, most notably the Results section is difficult to follow, and the introduction/discussion could use a more careful description of the theory behind the paper, and the implications the findings have for the theory.

Experimental design

The experimental design fits within the aims and scope of the journal. The research question is (mostly) well defined, relevant, and meaningful. The work was does with high technical standard, and the methods were (mostly) described with sufficient detail.

There were a few small questions remaining about the methods section, noted in the review.

Validity of the findings

The conclusions are well stated, and the authors shy away from too much speculation. I did not see access to the data, but I may have simply missed it.

Additional comments

Overall, it's a fine paper that should be published in PeerJ in some form. I have a handful of comments for the authors in my longer review, attached below.

Annotated reviews are not available for download in order to protect the identity of reviewers who chose to remain anonymous.

---

## Round 0.2 · Minor Revisions

Please follow carefully the reviewer's comments.

·

Basic reporting

No comments.

Experimental design

No comments.

Validity of the findings

No comments.

Additional comments

I highly appreciate for the authors exhaustive and reasonable reply. Most of them convince me except for below minor three points.

1) Please forgive me for my poor explanations.

My original comment:
L. 201
Here, change in the PSE and Weber fraction (WF) only were referred. Not only them, mean PSE and WF were calculated. Please describe them too.

Reply: Accordingly, we described the mean PSE and WF in the Results section:

My reply to the authors reply:
My intension was as follows: Around L 197 of the revised version, the authors wrote only "we calculated the change in the PSE" but not about the mean PSE (and now I recognized also about Weber fraction). But I think the mean PSE is more basic index than change in PSE. In fact, the mean PSE was shown in Fig. 1 & 2. When I see the Fig.1 at first, I was wondering why change in PSE was not shown here even though the author wrote they calculate change in PSE. So, my requirement was "please write that the authors calculated the mean PSE (and also Weber fraction) as well as change in PSE here, around L197". On the other hand, I agree with the addition of the mean values of them for the Fig.4 in the RESULTS section.


2) About the ANOVA of (treatment) x (time)
I agree with the addition of the ANOVA that is correspond to figures. However, in L236, "(Fig. 1D)" and "(Fig. 1E)" is remained. These figures do not show " For each group, the differences of PLs between baseline day and injection day (L235-236)". Please delete the term "(Fig. 1D)" and "(Fig. 1E)" here (not the Figures themselves!)


3) L 280
When an interaction is not significant, we cannot move onto the test of simple main effect. Here the interaction was not significant so that the description in the text and asterisks at 5.04-s in Fig2E should be delete. Alternately, the main effect of treatment was significant on the injection day. Therefor, an asterisk may be added between the legend of the symbols (between " --o-- Saline" and "--o—Formalin").


That's all

---

## Round 0.3 · accepted · Accept

Thank you for your careful responses to the reviewers' comments. Your results will add interesting information to the very few studies on that topic.